# The Significance of External Quality Assessment Schemes for Molecular Testing in Clinical Laboratories

**DOI:** 10.3390/cancers14153686

**Published:** 2022-07-28

**Authors:** Nele Laudus, Lynn Nijs, Inne Nauwelaers, Elisabeth M. C. Dequeker

**Affiliations:** Biomedical Quality Assurance Research Unit, Department of Public Health and Primary Care, KU Leuven, B-3000 Leuven, Belgium; nele.laudus@kuleuven.be (N.L.); lynn.nijs@kuleuven.be (L.N.); inne.nauwelaers@kuleuven.be (I.N.)

**Keywords:** external quality assessment, laboratory medicine, quality improvement

## Abstract

**Simple Summary:**

Patients and clinicians often rely on the outcome of laboratory tests, but can we really trust these test results? Good quality management is key for laboratories to guarantee reliable test results. This review focusses on external quality assessment (EQA) schemes which are a tool for laboratories to examine and improve the quality of their testing routines. In this review, an overview of the role and importance of EQA schemes for clinical laboratories is given, and different types of EQA schemes and EQA providers available on the market are discussed, as well as recent developments in the EQA landscape.

**Abstract:**

External quality assessment (EQA) schemes are a tool for clinical laboratories to evaluate and manage the quality of laboratory practice with the support of an independent party (i.e., an EQA provider). Depending on the context, there are different types of EQA schemes available, as well as various EQA providers, each with its own field of expertise. In this review, an overview of the general requirements for EQA schemes and EQA providers based on international guidelines is provided. The clinical and scientific value of these kinds of schemes for clinical laboratories, clinicians and patients are highlighted, in addition to the support EQA can provide to other types of laboratories, e.g., laboratories affiliated to biotech companies. Finally, recent developments and challenges in laboratory medicine and quality management, for example, the introduction of artificial intelligence in the laboratory and the shift to a more individual-approach instead of a laboratory-focused approach, are discussed. EQA schemes should represent current laboratory practice as much as possible, which poses the need for EQA providers to introduce latest laboratory innovations in their schemes and to apply up-to-date guidelines. By incorporating these state-of-the-art techniques, EQA aims to contribute to continuous learning.

## 1. Introduction

### Why Are Good EQA Schemes Important?

A great variety of high-tech laboratory tests and therapies have become available to patients with cancer in recent years. The latter are often expensive, leading to high health care costs as well as potential burden on the patient. Therefore, it is important that these testing methods and advanced treatments are only applied in appropriate cases. Laboratory biomarkers play a key role in patient selection due to their screening, diagnostic, prognostic and/or therapeutic values. In order to guarantee that reliable and accurate test results can be provided by a laboratory at any time, a good quality management system along with quality control are essential [1,2,3]. This includes regular enrollments in external quality assessment (EQA) schemes. The latter are defined by the World Health Organization (WHO) as “a system for objectively checking the laboratory’s performance using an external agency or facility” [4]. In this review, we will focus on molecular pathology testing. However, EQA schemes are available in many different fields and many of the same principles apply.

External agencies, also known as EQA providers, organize and coordinate EQA schemes aiming at more transparency amongst laboratories in contrary to ring studies or ring trials, which are often organized internally between laboratories without interference of an independent external body [5]. The term proficiency testing (PT) is often used interchangeably with EQA, although there are small distinctions. For example, EQA puts more emphasis on continuous improvement and education compared to PT, which attaches more importance to meeting regulatory requirements [6]. There are also differences between EQA providers. Some are affiliated to universities and perform research, for example the Biomedical Quality Assurance research unit (BQA, affiliated to the University of Leuven (KU Leuven)) and Nordic immunohistochemical Quality Control (NordiQC, affiliated to Aalborg University) [7,8]. Other EQA providers are affiliated to national/regional authorizing bodies and have a dual role (coordinating EQA schemes as well as providing official recognition of laboratories), e.g., the College of American Pathologists (CAP) [9]. For others, organizing EQA schemes is their core business, for example, European Molecular Genetics Quality Network (EMQN) [10]. EQA/PT providers can be accredited according to International Organization for Standardization (ISO)/International Electrotechnical Commission (IEC) 17043 to confirm that they work according to international standards [6,11,12]. ISO/IEC 17025 and ISO 15189 recommend that testing and diagnostic laboratories collaborate with ISO/IEC 17043-accredited EQA/PT providers [11,13,14].

In general, an EQA process consists of following steps: sample distribution, testing and reporting phase, assessment by experts (also called assessors) and appeal phase. Enrollment in EQA is one thing, but learning and improving based on the outcome, individual errors or those from other labs is another thing. It is not only laboratories can learn from an EQA scheme, but also the EQA providers themselves. EQA participation gives laboratories the opportunity to evaluate own performance over time as well as to compare own performance to that of peers. As such, laboratories can benchmark the performance of their daily testing method (laboratory-developed tests (LDTs) or commercial kits). In addition, EQA has proven its benefit for by creating the opportunity to set up collaborations with the EQA provider to investigate the use and performance of their kits. Previous EQA research has shown that EQA participation can reveal flaws in testing methods, for example essential differences between staining techniques [15,16]. In addition, EQA schemes, as well as ring studies, have had a supporting role in the development of best practice guidelines by highlighting pitfalls and common errors [17,18]. To further improve patient care, with a diversity of new testing methods, the role of EQA schemes and the lessons learned from these schemes will become very important.

## 2. How to Organize Different Types of EQA Schemes?

### 2.1. Different Types of EQA Schemes Have Different Purposes and Aim to Improve Different Phases/Parts of the Process

EQAs can be organized in many different contexts. EQAs are widely applicable and should be widely applied to aim for the best possible health care. Many biomarkers can be tested for different diseases with multiple methods and each method has its own advantages and challenges. For example, Anaplastic Lymphoma Kinase (ALK) alterations can be tested for using immunohistochemistry (IHC), fluorescent in situ hybridization (FISH), genetic testing on DNA or RNA level and ctDNA tests [19,20].

The purpose of the EQA scheme should be clear and be able to reach its goal in the way it is organized. For example, there are several EQA schemes set up recently to tackle an immediate problem to test for the presence of SARS-CoV-2. These EQA schemes have a clear goal: can test centers correctly identify the presence of the virus? Several types of tests were developed as quickly as possible and were important for the health of individual people as well as public health globally. Hasselmann et al. performed a first pilot scheme for SARS-CoV-2 testing which showed that IgG testing is more reliable than IgM testing, Ast et al. concluded from their quality assessment that certified tests produce more correct results than noncertified tests and Buchta et al. revealed that targeting specific genes of the virus result in more consistent results in their study [21,22,23]. Three different questions were tackled in these EQA schemes; hence, they are organized to address these points specifically. Without external quality control, these differences would not have been noticed as quickly. The purpose of finding the best tests to reliably test for the presence of the virus was fulfilled.

This example relates to a (currently) very common infection; however, rare diseases are tested for as well. In these cases, the purpose is not to urgently improve new tests; nevertheless, EQA is important to guarantee correct diagnoses to make sure proper treatment can be started as soon as possible (if available). The selection or “production” of samples for such EQA schemes are trickier with real cases not being available in big quantities. EQA schemes aim to provide the participant with samples that match real-life cases as closely as possible [24].

Since there are many different contexts and as there can be different goals for the same biomarker test, there are plenty of types EQA schemes aiming at pre-analytical, analytical and/or post-analytical phases of the testing process. Some focus on correct outcomes only, others want to improve technical performance of techniques, some want to focus on record keeping by focusing on elucidating reports to ensure clear and complete record keeping, and yet others focus on communication by aiming to remove ambiguity from result interpretation and conclusion.

Parts of the pre-clinical phase that can be tested (whether or not by questionnaire) are sample handling, sample storage, test ordering based on case history, sample preparation, error rates, sample rejection rates and causes, etc. [25]. For example, the influence of pre-analytical procedures and conditions on genomic DNA integrity in blood samples was studied by Malentacchi et al. and showed a significant difference in copy gene numbers as well as a high variability of gDNA integrity between laboratories [26,27].

The analytical phase is much better studied and involves sending samples to participants and assessing the results of laboratories. If the IHC stained slides are sent back to the EQA provider, the technical performance of staining can be assessed as well [16]. Many different methods can be investigated, even for the same biomarker as each method has their advantages and disadvantages [28]. For example, EQA schemes focusing on ALK can inquire about FISH methods only, IHC methods and the impact on interpretation, or a combination of all methods, including DNA/RNA testing [16,19,29,30].

### 2.2. EQA Providers Should Adhere to Several Guidelines to Guarantee High-Quality EQA Schemes

There are several guidelines, recommendations and an ISO norm dedicated to the proper set up of an EQA scheme. In this section, we will focus on (molecular) pathology EQA schemes and include recommendations from the field. The ISO/IEC 17043 standard is set up in a general sense as to be able to be applied each type of EQA scheme [11]. The requirements and recommendations are categorized in five main subjects and discussed by Dufraing et al. giving a consensus view from EQA providers with an international steering committee [12]. Further good practices are discussed by Tembuyser et al. for both diagnostic laboratories and EQA providers, as well the minimal requirements for clear and comprehensive clinical reports [31].

For example, when participants register for an EQA scheme, the purpose, number of samples and timeline should be clearly communicated by the EQA provider and a qualified team of medical and technical experts should be gathered [11,12,31]. Samples should be of good quality, relevant to the scheme (with reportable ranges that are relevant as well) and be selected by experts in the field which are deemed qualified by the EQA provider [11,12,31]. Note that it is not always possible to use real patient samples and artificial samples might not always reflect actuality, although this should be strived for. Next, participants have to submit the requested results and/or send their samples back. Exactly which input is requested should be clearly communicated to the participants [11]. The results are reviewed by assessors and according to procedures that were selected and drawn up before the start of the EQA scheme [11,12,31]. Finally, the participants will receive their results, which is the consensus that assessors agreed upon, and which the participants can appeal.

### 2.3. Risks That Might Influence the Quality of an EQA Scheme Should Be Addressed

EQA providers can face certain challenges during the organization of their schemes. Some of these challenges can influence the manner in which way the EQA scheme was planned and organized. For example, if there is little experience with EQA organization, it might be wise to perform internal audits to identify the shortcomings to be comply to the standard ISO/IEC 17043 and seek accreditation as soon as possible [11]. An EQA scheme should be organized in such a way that it mirrors routine laboratory practices as much as possible. As indicated above, samples should be properly selected, which can be achieved by working with a network of experts that are accredited. Tembuyser et al. have addressed several of these challenges [31].

Another challenge that lies beyond reach of EQA providers is the routine treatment of samples. The goal is to test the routine practice of laboratories, however, EQA samples are not always treated as such. It is a missed opportunity for participants if they do not treat their samples as they would any other sample as the routine testing is not revised in that case. The responsibility then lies with the participant and the EQA provider can do little more than emphasize that samples should be treated as routine samples as much as possible.

In general, an EQA provider with ISO/IEC 17043 accreditation is capable to assess possible risks and prepare to tackle challenges that might come up during the organization of an EQA scheme. ISO/IEC 17043 accreditation requires proper training of staff, clear procedures for selecting collaborators and selection of samples for each EQA scheme, sample preparation and storage procedures, assessment procedures regarding statistics, correct outcomes and methods, and procedures for data analysis [11]. Besides these steps of the process, there are also requirements to adhere to procedures regarding communication (e.g., clear instructions to participants, but also procedures for dealing with complaints) and confidentiality throughout the entire EQA process [11].

### 2.4. Types of EQA Providers and Schemes

EQA providers are responsible for the set-up, coordination and supervision of EQA schemes. There are many traits that characterize different providers. Some providers focus on one EQA scheme, while others organize many EQA schemes [32,33]. Many providers focus on national clinical laboratories, while other providers intend to reach a larger audience. Another trait and also an important one is the ISO/IEC 17043 accreditation status of the EQA provider.

#### 2.4.1. EQA Providers with ISO/IEC 17043 Accreditation

ISO/IEC 17043 accreditation ensures the correct organization and execution of the EQA schemes under accreditation, and the competence of the EQA provider [11]. However, accreditation of an EQA provider for one EQA scheme does not mean all EQA schemes by that provider are accredited. For each EQA scheme, it is decided whether it falls (completely) within the scope for audits or not. The EQA provider is only audited based on EQA schemes that are in scope and only those EQA schemes can be accredited. Furthermore, ISO/IEC 17043 details general requirements that can be applied for all types of EQA schemes and serves as a basis for more specific technical requirements that are different in each field [11]. Some examples of guidelines that are based on ISO/IEC 17043 accreditation, but go into more detail on technical requirements are by Dequeker et al. and Langerak et al. for testing of *CFTR* and suspected lymphoproliferations respectively [34,35].

ISO/IEC 17043 accreditation is awarded by the national accreditation body of the country where the EQA provider is based [36]. National accreditation bodies are listed on International Accreditation Forum (IAF) website (https://iaf.nu/en/accreditation-bodies/, accessed on 7 June 2022). Organizations that have received ISO/IEC 17043 accreditation by their national accreditation body are listed on the website of that accreditation body. It is important to note that providers do not have to be accredited to organize EQA schemes.

#### 2.4.2. Goals of EQA Providers

As mentioned before, EQA schemes can have different goals and the same applies for EQA providers. The goal of an EQA provider can be focused on organizing schemes to obtain the correct outcome of a test in national (reference) laboratories, while another goal of an EQA provider can be to aim at a larger intended population and to research which methods are used in the world and how these methods compare to one another. The feedback of each goal of an EQA scheme to the participating laboratory can therefore differ substantially.

For example, a Belgian governmental focused EQA provider, such as Sciensano (Table 1), might organize many EQA schemes with the goal of ensuring competence of clinical laboratories, adherence to policy requirements and improvement of quality of testing. Their intended population are Belgian clinical laboratories only and the feedback to laboratories includes whether the outcome of analysis was correct and possible areas of improvement [37].

Another EQA provider, CF Network, focuses on one specific gene, Cystic Fibrosis Transmembrane conductance Regulator (*CFTR*) [32]. They intend to improve quality of testing and perform research on longitudinal performance of genetic testing laboratories, interpretation of test outcomes and reporting of results (Table 1). The intended population is extended to any clinical laboratory testing for *CFTR* mutations worldwide. To perform research, more information is requested from the participants and more feedback is given at the end of the EQA scheme, for example on the reporting of detected mutations with variable outcomes [38].

One of the most important aspects of an EQA scheme is analyzing the outcome of tests. In molecular pathology, testing is not limited to detection of mutations as certain pathologies are caused by fusions of genes or clonal expansion of blood cells; hence, EQA providers also organize schemes for this specific purpose [39,40,41].

The type of samples used depends on the methods that will be inspected in the EQA scheme, hence EQA providers have to select their samples carefully. Ideally, a sample for an EQA scheme represents real-life samples with similar minimal abundance, variant allele frequencies that are relevant and a condition similar to the state they usually arrive in at hospital laboratories

For example, samples are sometimes processed first to preserve the sample as much as possible and testing only occurs afterwards on formalin-fixed, paraffin-embedded (FFPE) material. Similarly, DNA of samples can be extracted from bodily fluids in which case a tube with DNA is used for testing. Both options can be selected by an EQA provider, yet the decision on sample type should be based in reason.

If EQA providers focus on the diagnostic test specifically, sending out DNA extracted beforehand warrants similar DNA quality for testing between laboratories. However, a provider might want to include evaluation of the extraction process in the EQA scheme and send FFPE material from which laboratories need to extract DNA themselves. During analysis of results, these conditions need to be considered to be able to compare results between participants. The main goal of the EQA provider in a specific scheme needs to be kept in mind in each step of the EQA set-up.

#### 2.4.3. Research by EQA Providers Can Highlight Points of Concern

The outcome of different testing methods on the same sample should be the same; however, the sensitivity of two methods is not necessarily the same. This can be investigated by comparing participants testing the same sample with different methods.

Furthermore, intermediary results can be requested, as the methods for extraction can show a difference in extraction efficiency and yield less DNA, which, in turn, can suggest possible reasons for altered outcomes in samples with low variant allele frequencies.

Another point the EQA provider should take in mind in such cases is the level to which they will take into account certain criteria that might alter outcomes. If the methods used are known to the EQA provider and one of those methods is not sensitive enough to pick up a low variant allele frequency, this knowledge can be considered when analyzing results.

The extra information requested during research-oriented EQA schemes allows providers to inform participants in more detail on their results and their process of testing. Feedback can show that other laboratories with the same methods perform better or struggle with the same issues. The extra information can be valuable for EQA providers, but also ask for more detailed and exhaustive analysis of results. The EQA provider should be aware of the requirements for more complex analyses and ensure these requirements are met during the set-up of the EQA scheme.

## 3. Importance of EQA Schemes for Laboratories and for Clinical Practice

The overall goal of EQA schemes is to enhance patient safety and patient care by preventing laboratory errors by unveiling critical errors as well as areas of improvement in laboratory testing. EQA helps laboratories to monitor own performance over time and to benchmark own performance to that of peers. Marking criteria used during EQA assessment should be clearly reported and substantiated by the experts. According to ISO/IEC 17043 following measurements are recommended: overall performance against prior expectations, inter- and intra-participant variation, variation between methods or procedures, educational feedback and advice, general remarks, etc. [11]. Feedback obtained during EQA participation can be translated by the laboratories to corrective and preventative actions (CAPAs) in practice. Miller et al. made a classification of distinct problems that can be discovered during EQA, including clerical errors, methodological problems, equipment problems, technical problems caused by personnel errors or a problem with the PT material [42]. It is the laboratory’s responsibility to actively improve their practice. If participants do not agree with their scoring or do not understand certain feedback, they can submit an appeal to request further explanation on their outcome or (for some providers) ask for help with implementation of CAPAs or improvement measures in their laboratory. In addition, regular enrollment in EQA schemes is obliged in order to be accredited as a testing/medical laboratory according to ISO/IEC 17025 or ISO 15189 [13,14]. In countries like France, laboratories dealing with human samples for diagnostic purposes are to be ISO 15189-accredited [13]. Previous retrospective studies regarding external quality control in genetics, as well as molecular pathology, has shown improvement in laboratory performance amongst laboratories that participated regularly in EQA schemes, highlighting the educational role of EQA [1,38,43,44,45,46]. Research has shown that it is beneficial for laboratories to start with EQA in an early phase after introducing new biomarkers in the laboratory [45]. Nevertheless, we have to be aware that besides the educational aspect of EQA, other factors can contribute to these results. For example, familiarity with the scheme, participation bias and evolution of scheme organization can influence performance [47].

## 4. The Scientific Value of EQA Schemes and Importance for Industry: What Can EQA Schemes Learn Us?

### 4.1. Insights from EQA Schemes: Performance of Laboratories

In addition to individual performance assessment of laboratories, EQA schemes provide a unique insight contributing to scientific questions by comparing large-scale data from international laboratories.

For example, EQA schemes in the field of molecular pathology have uncovered several critical errors during all phases of the test process. Generally, laboratories are responsible for ensuring proper training and execution of the different steps of the analysis. However, the extent and the manner in which training for particular steps is fulfilled can vary between laboratories as well as between pathologists [45].

The samples that are distributed during EQA schemes need to fulfill specific conditions in order for the participant to receive comparable samples [11]. However, the outcomes of an analysis are still dependent on pathologist’s interpretation. For example, selected tumor samples that are distributed have a specific minimum amount of tumor cells and pathologists should be able to interpret this percentage accurately [48]. Nonetheless, during the pre-analytical phase, a large variation in the estimation of tumor cellularity in tumor samples has been uncovered [48,49]. Due to the influence of over- or under-estimations on the test outcome, a need for harmonization is apparent. In part, training plays an important role [50]. To implement this, large-scale data on these estimations are needed in order to define between-observer concordances and determine bias-introducing factors in the test process.

Similarly, a comparison of three RING trials conducted by the Tumor-Infiltrating Lymphocytes (TILs) Working Group assessing stromal TILs scoring (sTILs) has highlighted the main pitfalls and errors during this determination. Intraclass correlation coefficients (ICC) between the three ring studies ranged from 0.70 (0.62–0.78) to 0.89 (0.85–0.92) and were influenced by hurdles such as inflammatory cells, tumor boundary and other factors [17]. By comparing these ring trials, it was shown that variation between TILs estimation could be addressed by (1) providing reference images with a predefined TIL percentage for comparison, and (2) evaluation of multiple tumor areas which avoids intratumoral heterogeneity and random errors [51].

In general, with regard to the analytical phase, accreditation enhances a laboratory’s performance, meaning that they have overall fewer analysis errors, deviating results or technical errors than non-accredited laboratories [45,52]. These results demonstrate the importance of EQA participation in reaching a successful performance. Since accreditation of medical laboratories is not obligatory in every country, participation in EQA schemes is even more important to reach satisfactory test quality. Furthermore, when implementing a new biomarker, accreditation and successive EQA participation result in a swift implementation into routine clinical practice, possibly relating to the implementation procedure [45,53].

Longitudinal data from EQA schemes have shown that the implementation of an alternative predictive molecular test in a laboratory has positive effects on false-negatives or erroneous results, but introduce more technical failures indicating the importance of adequate training before implementation in routine practice [54].

In terms of sample processing, the immunohistochemical staining performance for several biomarkers (such as Programmed Death Ligand-1 (PD-L1), ALK and ROS1) is also varying and has been demonstrated to be positively influenced by multiple EQA participations, most likely due to comparison to peers and post-EQA individual feedback [16,55]. Overall, the use of commercial kits results in a better staining performance compared to LDTs [16,56]. Nonetheless, essential for an optimal staining score is a validated protocol obtaining a good staining quality, while paying attention to interpretation competence of the pathologist to reduce inter-individual variation and avoiding misclassification/misinterpretation [15,16,57,58]. In addition, accreditation also had a positive impact on staining performance and interpretation for these biomarkers [55].

Post-analytically, variant description must be correct and according to the Human Genome Variation Society (HGVS) guidelines [59,60]. In a study by Tack et al. performed in 2016, results from four different EQA providers (European Society of Pathology (ESP), EMQN, UK NEQAS and the French national Gen&Tiss EQA scheme) were compared to determine common variant nomenclature errors. Areas of problems included, among others, clerical errors, use of traditional nomenclature instead of HGVS nomenclature and omission of p. or c. [10,41,61]. However, the provision of detailed feedback has been shown to be useful to improve during subsequent participation [62].

Thus, research has demonstrated that participation in EQA schemes is a valuable tool for improvement of test quality in laboratories, as well as a means for feedback on a personal level to improve interpretation and reporting. In addition, the evidence developed in this field has led to the development of guidelines and recommendations to advise medical laboratories on participation in external quality assessment programs and on the phases in the testing process (e.g., analysis of markers); interpretation and reporting; HGNC guidelines; and HGVS guidelines [28,35,60,63,64,65,66,67,68,69].

### 4.2. Insights for Industry: Performance of Available Kits and Test Methods

Data from EQA schemes also uncover information about the sensitivity of certain predictive methods. In 2020, research of Keppens et al. showed that for the correct detection of secondary *EGFR* c.2369C>T p.(Thr790Met) in non-small cell lung cancer (NSCLC), next-generation sequencing-based techniques (NGS) and non-NGS commercial kits were superior to in-house non-next-generation sequencing techniques [54].

The detection of variants and subsequent administration of targeted therapy is complicated further by the variability in companion diagnostics, only allowing reimbursement of specific therapies when a specific diagnostic assay is used as is the case for *PD-L1* [70]. In this context, several articles reported significant differences during ring trials between the percentage of acceptable or correct results achieved by laboratories using LDTs and U.S. Food and Drug Administration-approved companion diagnostics (FDA-CDx) [71,72]. Therefore, outcomes from these studies remain important to determine whether commercial kits or CDx perform well and if these are properly used, or whether they should be updated to raise sensitivity and include certain (rare) variants.

## 5. Challenges and Future of EQA Schemes

EQAs, as they exist now, can be used to increase the study power of clinical trials. When introducing new biomarkers in clinical trials, not only the biomarker performance characteristics will be evaluated, but also the interpretation of the biomarker presence in tumor tissue by the pathologist can be assessed. By doing this, between-laboratory variability in performance and interpretation can be minimized, while increasing the study power [73].

However, EQA schemes don’t have a standard format and can evolve to a more innovative set-up. Whereas conventional EQA schemes assess performance on laboratory level, the current issues, such as differences in staining performance and interpretation, ask for a shift towards assessment of individual participation (Figure 1) [74]. Individual-based EQA schemes provide the possibility to uncover individual-related problems and aim at improvement of practices by training [75]. This type of EQA scheme might also be useful in the training of pathologists and laboratory personnel to perform optimally in daily practice. As a result of feedback and comparison between peers, interindividual variation in performance would also be diminished, which would enhance the comparability between similar studies.

An example of an individual-based EQA scheme is the scheme that will be organized by the international TILs working group in collaboration with a group of experts, to optimize TIL and PD-L1 estimation in breast cancer [8,76]. A digital training and subsequent EQA scheme will be organized to which all pathologists have free access to register. Different educational applications are available for the training phase, including a guideline, a tutorial and a training tool enabling pathologists to score pathology images. After training, pathologists will have the opportunity to improve their skills by scoring TILs and PD-L1 on real-life digital pathology samples. TILs are to be evaluated on Hematoxylin and Eosin (H&E)-stained slides and different PD-L1-stained IHC slides (e.g., Ventana 142, Ventana 263, Agilent 22C3) will be provided for the PD-L1 part of this scheme. Individual feedback concerning performance enables improvement in daily practice, contributing to a more harmonized determination of prognosis and therapy response prediction.

Digital EQA schemes, such as the example provided here, use strictly digital cases. Digital pathology and its inclusion in these more innovative types of schemes have the advantage that any differences due to intrinsic sample variation and sample processing methods are circumvented. As participants receive identical images, the method-related differences are eliminated, thus allowing for an evaluation of interpretation [77].

The use of digital cases also circumvents the issue of not wanting to use artificial samples, as these might differ from routine cases, as well as not having a lot of material from real samples. With limited tissue, all pathologists can still assess real samples and interpret actual cases with all the complications or pitfalls that come with it.

In addition, the latest advancements in machine learning and artificial intelligence in pathology result in a more precise detection, diagnosis, prognosis and therapy prediction, thus rendering inter-individual variability to a minimum [78,79,80]. Where automated analysis of these digitized images would enhance harmonization in interpretation, there are still a few obstacles for effective implementation, including a lack of legal reference frames, ethical considerations, proper validation of these tools and the differences in outcomes generated by different tools [80,81,82].

Since there are many different providers with different goals for each EQA, this can also lead to different standards of achievement. Harmonization between EQA providers is a big challenge and there are many other obstacles ahead [83]. However, total harmonization might not be feasible or wanted if we want to keep benefiting from of the different purposes of EQA schemes.

## 6. Conclusions

The aim of EQA is to assist laboratories in assessing testing performance and to contribute to continuous learning. Various EQA schemes are available for laboratories, each with their own characteristics and purpose. Laboratory medicine is a fast-evolving field, requiring EQA providers to reflect new evolutions in their schemes. This is not only important for laboratories, but also industry and research institutes can benefit from studying the results of EQA schemes. Furthermore, EQA providers should keep evolving with the field they serve and tackle the challenges that they encounter to keep addressing their goals.

## Figures and Tables

**Figure 1 cancers-14-03686-f001:**
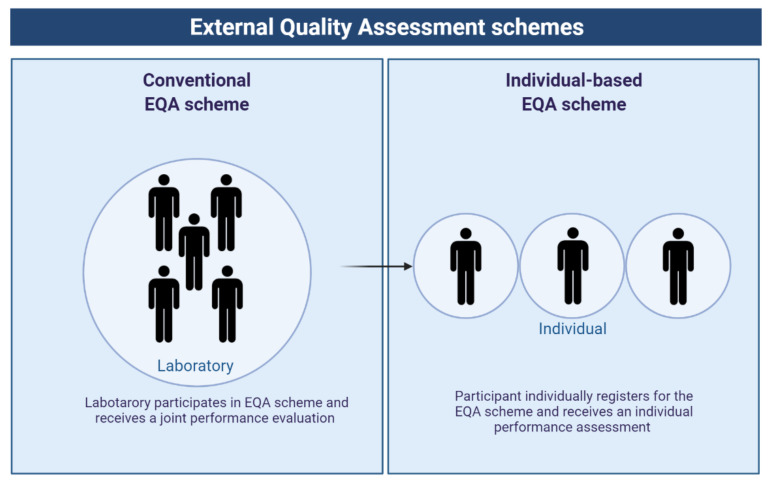
Shifting format of EQA schemes from laboratory-level assessment to individual-based assessment to fit the purpose of current research questions. Abbreviations: EQA, External Quality Assessment. Figure created with BioRender.com.

**Table 1 cancers-14-03686-t001:** Overview of types of EQA providers and their characteristics (situation in 2021). This table serves as an example and is not an exhaustive list.

EQA Provider	National versus International	Country of Headquarters	Use	Goals of EQA Types Organized by this Provider	Type of Sample Used	Genes Tested	ISO 17043 Accredited	Methods Assessed	References
ESP	International: worldwide	Belgium	Expertise & Research	Genotyping Technical assessment Reporting Interpretation	Patient samples (FFPE)	*PD-L1* in four types of cancer *KRAS, NRAS, BRAF* in CRC Multiple genes in NSCLC	Unclear	IHC, FISH, DRT, DNA analysis (PCR and NGS)	https://www.esp-pathology.org/
EMQN	International: worldwide	United Kingdom	Expertise	Genotyping Reporting Interpretation	Many types of samples	Many genes in many types of pathology	Some EQA schemes	Many methods assessed	https://www.emqn.org/eqa-scheme-catalogue/
CF Network	International: worldwide	Belgium	Expertise & research	Genotyping Reporting Interpretation	Patient sample (DNA)	*CFTR*	Yes	DNA analysis (PCR and NGS)	http://cf.eqascheme.org/
EuroClonality	International: worldwide	The Netherlands	Expertise	Genotyping Interpretation	Patient sample (DNA)	IG and TR clonal expansion	Yes	DNA analysis (PCR)	https://euroclonality.org/eqa-scheme
UK NEQAS	International: worldwide	United Kingdom	Expertise	Genotyping Technical assessment Reporting Interpretation	Many types of samples	Many genes in many types of pathology	Some EQA schemes	Many methods assessed	https://ukneqas.org.uk/programmes/
RCPAQAP	International: worldwide	Australia	Expertise & research	Genotyping Technical assessment Reporting Interpretation	Many types of samples	Many genes in many types of pathology	Some EQA schemes	Many methods assessed	https://rcpaqap.com.au/products/
Sciensano	National: Belgium	Belgium	Policy	Genotyping Reporting Interpretation	Digital and artificial samples (DNA)	Many genes in many types of pathology	Yes	Many methods assessed	https://www.sciensano.be/en/about-sciensano/sciensanos-organogram/quality-laboratories/external-quality-assessment#want-to-know-more- and https://www.wiv-isp.be/QML/index_nl.htm
Gen&Tiss	National: France	France	Policy & research	Genotyping Technical assessment Reporting Interpretation	Patient and artificial samples (DNA, ctDNA)	*EGFR*, *KRAS*, *BRAF*, *NRAS*, *PIK3CA*, MSI, *ERBB2*, *KIT* in different types of cancers	Yes	DNA analysis (PCR and NGS) and ctDNA analysis	http://www.genetiss.org/
CAP	National: United States of America	United States of America	Expertise	Genotyping Reporting Interpretation	Many types of samples	Many genes in many types of pathology	Yes	Many methods assessed	https://www.cap.org/laboratory-improvement/proficiency-testing
CIQC (split in CPQA-AQCP and CBQA-PCAB)	National: Canada	Canada	Expertise, policy & research	Genotyping Technical assessment Reporting Interpretation	Origin unclear (DNA, FFPE)	*BRCA, EVER, P16, NTRK, PD-L1, HER2* and more	No	IHC, FISH, DNA analysis (PCR and NGS)	https://www.cpqa.ca/ and www.cbqa.ca
NCCL	National: China	China	Policy	Genotyping Reporting	Unclear	Unclear	Yes	Many methods assessed	https://www.nccl.org.cn/planEn

Abbreviations: BRCA, Breast Cancer gene; CAP, College of American Pathologists; CBQA-PCAB, Canadian Biomarker Quality Assurance-Programme Canadien d’assurance de la qualité des biomarqueurs; CFTR, Cystic Fibrosis Transmembrane conductance Regulator; CIQC, Canadian Immunohistochemistry Quality Control; CPQA-AQCP, Canadian Pathology Quality Assurance–Assurance qualité canadienne; CRC, Colorectal cancer; ctDNA, Circulating tumor DNA; DRT, Digital Readout Test; EGFR, Epidermal Growth Factor Receptor; EMQN, European Molecular Genetics Quality Network; EQA, External Quality Assessment; ERBB2, Erb-B2 Receptor Tyrosine Kinase 2; FFPE, Formalin-fixed, Paraffin-embedded; FISH, Fluorescent In Situ Hybridization; HER2, Human Epidermal growth factor Receptor 2; GTPase, Guanosine Triphosphatase; IG, Immunoglobulin; IHC, Immunohistochemistry; KRAS, Kirsten rat sarcoma viral oncogene homolog; MSI, Microsatellite instability; NCCL, National Center for Clinical Laboratories; NGS, Next-Generation Sequencing; NRAS, Neuroblastoma RAS viral oncogene homologue; NSCLC, Non-Small Cell Lung Cancer; NTRK, Neurotrophic tyrosine receptor kinase; PCR, Polymerase Chain Reaction; PD-L1, Programmed Death Ligand-1; PIK3CA, Phosphatidylinositol-4,5-Bisphosphate 3-Kinase Catalytic Subunit Alpha; RCPAQAP, Royal College of Pathologists of Australasia Quality Assurance Programs; TR, T-cell Receptor; UK NEQAS, United Kingdom National External Quality Assessment Service. Accessed 7 June 2022.

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
