# Peer review of "The Significance of External Quality Assessment Schemes for Molecular Testing in Clinical Laboratories"

_cancers, 2022, doi:10.3390/cancers14153686_

Round 1

Reviewer 1 Report

The article is a very well written review of current practices in EQA and  describes the role and usefulness of EQA.

I have one major comment:

1. The examples given in and references from this article show that it is written from the framework of EQAs in molecular pathology. It makes sometimes reference EQAs in other domains as well, and this may be confusing for the reader.
For example:
- Title:  the concept of biomarker testing in clinical laboratories is broader than the testing that the article deals with.
- Section 2.1: This section aims at describing EQAs in the clinical laboratories in the general context, but does not mention the classification made by Miller et al. for testing in the field of biochemistry (Miller, W. Greg, et al. "Proficiency testing/external quality assessment: current challenges and future directions." Clinical chemistry 57.12 (2011): 1670-1680).
- Table 1 only lists EQAs in molecular pathology.
- From section 2.2, teh article states clearly that it focusses on (molecular pathology).

I recommend to change the title and change the text in the beginning (up to section 2.1) to emphasize that this article deals mainly with (molecular) pathology testing.

In addition, I have some minor comments:
2. Table 1:  this list is not complete.  For example, Labquality (www.labquality.fi) and Equalis (www.equalis.se) also organize EQA in the field of molecular pathology testing.
3. Table 1:  Sciensano uses more than only digital slides.  See for example Delcourt, Thomas, et al. "Ngs for (Hemato-) oncology in belgium: Evaluation of laboratory performance and feasibility of a national external quality assessment program." Cancers 12.11 (2020): 3180.
4. The last section, dealing with challenges and future of EQA schemes, does not cover all the challenges of EQA schemes.  The authors mentioned for example already the use of artificial samples earlier in the text; they do not write about this challenge in this section. I also recommend the authors to read the following manuscript: Badrick, Tony, and Anne Stavelin. "Harmonising EQA schemes the next frontier: challenging the status quo." Clinical Chemistry and Laboratory Medicine (CCLM) 58.11 (2020): 1795-1797.

Although it does not deal with EQAs for molecular pathology, I believe that joining data bases across EQA providers is a promising challenge for all EQAs in the medical field.

Author Response

Dear reviewer,

Thank you for thoroughly reviewing our manuscript. Our reply to your comments:

  1. We changed the title to “The significance of external quality assessment schemes for molecular testing in clinical laboratories” to stress the focus on EQA for molecular testing in this manuscript. In the introduction, we have added a sentence contextualizing the main focus of this manuscript.
  2. We kindly refer you to the adapted table 1 in the enclosed manuscript. In addition, we have indicated that the table is not exhaustive.
  3. Thank you for your feedback. We included digital and artificial samples provided by Sciensano in table 1.
  4. We agree that not all challenges are discussed in this manuscript (due to the limited space of this review). As you have mentioned, harmonization between providers, including joint databases, could be beneficial for EQA participants as well providers. However, each provider has its own unique way of addressing matters, which can also be beneficial for the participant to get familiarized with different approaches.

We hope this answers your questions/remarks and we love to hear your feedback.

Reviewer 2 Report

The overall contents of the EQA scheme are very well organized. I think it would be a good reference for the EQA scheme.

1.  Volume

The overall length of the thesis seems a bit long. Please reduce the length of the thesis by omitting repetitive and non-essential parts.

2.  Title

A.  There seems to be a slight difference between the title and content of this paper. The overall content is about EQA scheme and types, etc. Instead of ‘the significance of EQA’ in the title, how about changing it to such as the general requirements for EQA schemes?

B.  How about changing to molecular tests rather than the term biomarker?

C.  If the title is not limited to molecular tests, it would be better to delete the term biomarker.

3.  Content

A.  You explained the importance of the EQA system well. However, the most important step among the reasons why the EQA system is needed is the feedback on the EQA results.

Therefore, please briefly mention the criteria for judging the EQA results, accreditation or certification rules, etc. Also, please mention the importance of countermeasures for how each test institution feedbacks EQA results.

B.  I would like to change the shape of Table 1 to make it easier to see.

Author Response

Dear reviewer,

Thank you for thoroughly reviewing our manuscript. Please find below our reply to your comments:

  1. We have omitted some repetitive data, e.g., in the introduction section.
  2. We changed the title to “The significance of external quality assessment schemes for molecular testing in clinical laboratories” to stress the focus on EQA for molecular testing in this manuscript. Thank you for your suggestion.
  3. We have added a section on evaluation of performance according to ISO/IEC 17043 under section 3. We also briefly discuss the translation of EQA results in practice, e.g., by using CAPAs.
  4. We apologize for the format of table 1. We changed the format to landscape to make it more readable.

We hope this answers your questions/remarks and we love to hear your feedback.

Reviewer 3 Report

This review is extremely interesting, and represents a valid overview of the actual significance of EQA schemes for biomarker testing in clinical laboratories. 

The present work is very well structured and also well summarizes the actual international situation regarding EQA organizations, providers and significance.

I strongly reccomend this paper for pubblication.

Author Response

Dear reviewer,

Thank you for your feedback.